# COMFORT ZONE: A VICINAL DISTRIBUTION FOR REGRESSION PROBLEMS

## ABSTRACT

Domain-dependent data augmentation methods generate artificial samples using transformations suited for the underlying data domain, such as rotations on images and time warping on time series data. However, *domain-independent* approaches, e.g. `mixup`, are applicable to various data modalities, and as such they are general and versatile. While `mixup`-based techniques are used extensively in classification problems, their effect on regression tasks is somewhat less explored. To bridge this gap, we study the problem of domain-independent augmentation for regression, and we introduce `comfort-zone`: a new data-driven domain-independent data augmentation method. Essentially, our approach samples new examples from the tangent planes of the train distribution. Augmenting data in this way aligns with the network tendency towards capturing the dominant features of its input signals. Evaluating `comfort-zone` on regression and time series forecasting benchmarks, we show that it improves the generalization of several neural architectures. We also find that `mixup` and noise injection are less effective in comparison to `comfort-zone`.

## 1 INTRODUCTION

Classification and regression problems primarily differ in their output's domain. In classification, we have a finite set of labels, whereas in regression, the range is an infinite set of quantities—either discrete or continuous (Goodfellow et al., 2016). In classical work (Devroye et al., 2013), classification is argued to be "easier" than regression, but more generally, it is agreed by many that classification and regression problems should be treated differently (Muthukumar et al., 2021). Particularly, the differences between classification and regression are actively explored in the context of regularization. Regularizing neural networks to improve their performance on new samples has received a lot of attention in the past few years. One of the main reasons for this increased interest is that most of the recent successful neural models are *overparameterized*. Namely, the amount of learnable parameters is significantly larger than the number of available training samples (Allen-Zhu et al., 2019a;b), and thus regularization is often necessary to alleviate overfitting issues. Recent studies on overparameterized linear models identify conditions under which overfitting is "benign" in regression (Bartlett et al., 2020), and uncover the relationship between the choice of loss functions in classification and regression tasks (Muthukumar et al., 2021). Still, the regularization of deep neural regression networks is not well understood.

In this work, we focus on a common regularization approach known as Data Augmentation (DA) in which data samples are artificially generated and used during training. In general, DA techniques can be categorized into domain-dependent (DD) methods and domain-independent (DI) approaches. The former techniques are specific for a certain data modality such as images, whereas the latter methods typically do not depend on the data modality. Numerous DD- and DI-DA approaches are available for classification tasks (Shorten & Khoshgoftaar, 2019; Shorten et al., 2021), and many of them consistently improve over non-augmented models. Unfortunately, DI-DA for regression problems is a significantly less explored topic. Recent works on linear models study the connection between the DA policy and optimization (Hanin & Sun, 2021), as well as the generalization effects of linear DA transformations (Wu et al., 2020). We contribute to this line of work by proposing and analyzing a new domain-independent data augmentation method for nonlinear deep regression, and by extensively evaluating our approach in comparison to existing baselines.

Many strong data augmentation methods were proposed in the past few years. Particularly relevant to our study is the family of `mixup`-based techniques that are commonly used in classification

applications. The original method, `mixup` (Zhang et al., 2017), produces convex combinations of training samples, promoting linear behavior for in-between samples. The method is domain-independent and data-agnostic, and it was shown to solve the Vicinal Risk Minimization (VRM) problem instead of the usual Empirical Risk Minimization (ERM) problem. In comparison, our approach is domain-independent and *data-driven*, and it can also be viewed as solving a VRM problem. Through extensive evaluations, we will show that `mixup` and noise injection are less effective for regression.

**Contribution.** Challenged by the differences between classification and regression and motivated by the success of domain-independent methods such as `mixup`, we propose a simple, domain-independent and data-driven DA routine, termed `comfort-zone` (Sec. 3). Let $X, Y$ be the input and output mini-batch tensors, respectively, and let $Z_l = g_l(X)$ be the hidden representation at layer $l$ (Verma et al., 2019). Essentially, our method produces new training samples $Z_l(\lambda), Y(\lambda)$ from the given ones by scaling their small singular values by a random $\lambda \in [0, 1]$. At its core, `comfort-zone` incorporates into training the assumption that data with similar dominant components of the train set should be treated as true samples. We offer two simple implementations of `comfort-zone`; a non-differentiable approach that can be used for input-level application, and a fully differentiable pipeline which is applicable to any layer (App. A).

We analyze `comfort-zone` using perturbation theory, and we introduce its associated vicinal risk minimization (Sec. 4). Our experimental evaluation focuses on benchmark regression tasks (Sec. 5.1), and on time series forecasting tasks with small and large datasets (Sec. 5.2). The results show that `comfort-zone` improves the test error on several neural architectures and datasets, and in comparison to other DA baselines. We offer a potential explanation to the success of our method (Sec. 3, App. B). Finally, an ablation study is performed, justifying our design choices (App. C).

## 2    RELATED WORK

Deep neural networks regularization is an established research topic with several existing works (Goodfellow et al., 2016). Common regularization approaches include weight decay, dropout (Srivastava et al., 2014), batch normalization (Ioffe & Szegedy, 2015), and data augmentation (DA). In what follows, we categorize DA techniques to be either domain-dependent or domain-independent. Domain-dependent DA was shown to be effective for, e.g., image data (LeCun et al., 1998) and audio signals (Park et al., 2019), among other domains. However, adapting these methods to new data domains is typically challenging and often infeasible. In the past few years, an increased interest has been drawn to domain-independent DA methods, allowing to regularize neural networks when only basic data assumptions are allowed. We focus in what follows on **domain-independent** techniques that were proposed in the context of classification and regression problems.

**DA for classification.** Recently, Zhang et al. (2017) proposed to convex mixing of input samples as well as one-hot output labels during training. The new training procedure, named `mixup`, minimizes the Vicinal Risk Minimization (VRM) problem instead of the typical Empirical Risk Minimization (ERM). Many extensions of `mixup` were proposed, including mixing latent features (Verma et al., 2019), same-class mixing (DeVries & Taylor, 2017), among others (Guo et al., 2019; Hendrycks et al., 2019; Yun et al., 2019; Berthelot et al., 2019; Greenewald et al., 2021; Lim et al., 2021).

**DA for regression.** Significantly less attention has been drawn to designing domain-independent data augmentation for regression tasks. A recent survey (Wen et al., 2020) on DA for time series data lists a few basic augmentation methods including noise injection. Incorporating noise in the data can be used for regression tasks, and it can also be incorporated into other DA methods such as ours. Dubost et al. (2019) propose to recombine samples for regression tasks with countable outputs, and thus their method can not be directly extended to the uncountable regime. Recently, `mixRL` (Hwang & Whang, 2021) developed a meta learning framework based on reinforcement learning for mixing samples in their neighborhood.

```
def scale_down(Z, lam, rho):
  U, s, Vt = numpy.linalg.svd(Z)
  cumperc = numpy.cumsum(s) / numpy.sum(s)
  s *= numpy.where(cumperc > rho, lam, 1.0)
  Z = U @ numpy.diag(s) @ Vt

# X, Y are in batch x feats
for (X, Y) in loader:
  lam = numpy.random.beta(alpha, alpha)
  Z = numpy.concatenate((X,Y), axis=1)
  Z = scale_down(Z, lam, rho)
  X, Y = Z[:, :m], Z[:, m:]
  optimizer.zero_grad()
  loss(net(X), Y).backward()
  optimizer.step()
```

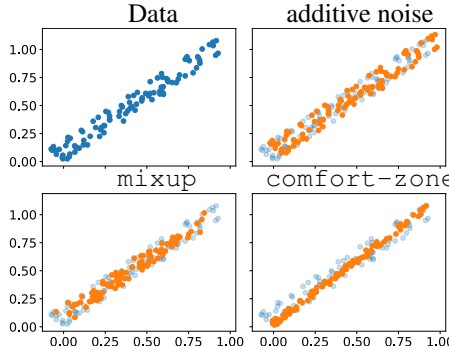

Figure 1: We show the pseudocode for comfort-zone at the input level, $l = 0$ (left). We demonstrate the effect of a few DA methods on 2D data whose intrinsic dimension is one (right).

## 3 COMFORT ZONE

A learning task is typically described as a function which maps inputs to outputs. In this view, a learning model is approximating that function using e.g., a neural network, and it is formulated via $f : \mathcal{X} \to \mathcal{Y}$, denoting the input and output domains by $\mathcal{X}$ and $\mathcal{Y}$, respectively. A regression problem is such that the output domain is (un)countable, e.g., $\mathcal{Y} \subset \mathbb{N}^m$ or $\mathcal{Y} \subset \mathbb{R}^m$. For simplicity, we consider the setting $\mathcal{X}, \mathcal{Y} \subset \mathbb{R}^m$, but our method is applicable to other cases. During training, the learning model is provided with a training set $\mathcal{D} = \{(x_i, y_i)\}_{i=1}^n$, sampled from $(x_i, y_i) \sim \mathcal{P}$. Our method extends the training distribution by producing a new training set as we describe below.

To generate new samples, we consider the singular value decomposition (SVD) of a matrix $A \in \mathbb{R}^{q \times r}, q \geq r$ which is given by $A = USV^T$. The matrices $U, V$ are orthogonal, and $S$ is a diagonal matrix whose main diagonal consists of the singular values ordered by $\sigma_1 \geq \sigma_2 \geq \cdots \geq \sigma_r \geq 0$. SVD is intimately related to principal component analysis (PCA) which in turn is heavily studied in manifold learning and dimensionality reduction (Ma & Fu, 2012). It is well-known that the best rank $k$ approximation of $A$ is given by omitting the last $(r - k)$ singular values, i.e., $A_k = \sum_{j=1}^k \sigma_j u_j v_j^T$ (Eckart & Young, 1936). The matrix $A_k$ preserves the $(k)$ dominant components in $A$, and it discards the rest. The key insight in our approach is that scaling down the small singular values should yield training samples that are in close proximity to the true distribution of the data $\mathcal{P}$.

Let the input and output mini-batch tensors $X \in \mathbb{R}^{b \times m}$ and $Y \in \mathbb{R}^{b \times m}$, respectively, where w.l.o.g $b \geq 2m$ is the batch size. We denote the network by $f(X) = f_l(g_l(X))$, $Z_l := g_l(X)$ where $g_l$ maps inputs to latent representations $Z_l$ at layer $l \in [0, L]$, and $f_l$ maps latent vectors to outputs (Verma et al., 2019). Let $\lambda \sim \text{Beta}(\alpha, \alpha)$ for $\alpha \in (0, \infty)$, and $k \in [1, 2m]$ be the index of the singular value after which we scale down. Then, the new artificial samples $Z_l(\lambda, k), Y(\lambda, k)$ are defined via

$$A := [Z_l,\ Y] = USV^T \in \mathbb{R}^{b \times 2m}\ ,$$
$$A(\lambda,\ k) := [Z_l(\lambda,\ k),\ Y(\lambda,\ k)] = US(\lambda,\ k)V^T\ ,$$

where $[\cdot, \cdot]$ concatenates along columns, and $S(\lambda,\ k)$ is the diagonal matrix of scaled down singular values. Namely, we compute $S(\lambda,\ k) = \text{diag}(\sigma_1, \ldots, \sigma_k \mid \lambda\sigma_{k+1}, \ldots, \lambda\sigma_{2m})$. The value $k$ depends on the hyper-parameter $\rho \in [0, 1]$ that represents the "amount" of signal to keep unchanged, i.e.,

$$k = \arg\max_{\tilde{k}} \sum_{j=1}^{\tilde{k}} \sigma_j / \sum_j \sigma_j \leq \rho\ .$$

Similar to mixup (Zhang et al., 2017), our method recovers the original dataset $\mathcal{D}$ as $\alpha \to 0, \forall \rho$.

The loss function associated with comfort-zone is

$$\mathcal{L}(f) = \mathbb{E}_{(X,Y)} \mathbb{E}_\lambda \mathbb{E}_l\, c\left((f_l, 1) \circ \chi(g_l(X), Y; \lambda, k))\right)\ ,$$
$$\text{s.t.}\quad (X, Y) \sim \mathcal{D},\ \lambda \sim \text{Beta}(\alpha, \alpha),\ l \sim [0, L]\ ,$$

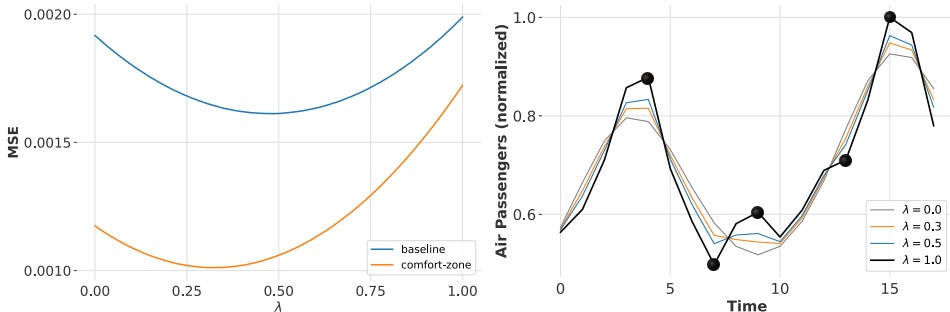

Figure 2: Evaluating a non-augmented model and a model trained with `comfort-zone` on train data whose small singular values are scaled down for different values of $\lambda$ (left). We show on the right panel an example of a time series sample (black), and its modifications using $\lambda = 0.5$ (blue), $\lambda = 0.3$ (orange), and $\lambda = 0.0$ (gray).

where $c : \mathbb{R}^{b \times m} \times \mathbb{R}^{b \times m} \to [0, \infty)$ is a cost function, typically mean squared error (MSE). The transformation $\chi$ takes a pair of tensors $g_l(X), Y$, and it scales down the last $(2m - k)$ singular values of their concatenation by $\lambda$. A key attribute of `comfort-zone` is that it is fully *differentiable* since the singular value decomposition can be backpropagated (Ionescu et al., 2015). Importantly, when `comfort-zone` is applied only at the input level ($l = 0$), a straightforward non-differentiable implementation is sufficient. We provide an example pyTorch pseudocode for input-level `comfort-zone` in Fig. 1 (left), and we discuss in App. A a potential differentiable implementation. The computational complexity of `comfort-zone` is governed by SVD calculation which has a complexity of SVD is $\mathcal{O}(\min(qr^2, rq^2))$ for an $q \times r$ matrix. In comparison, `mixup` samples a scalar from a random distribution, and it linearly blends two samples, whereas additive noise samples points from a random distribution. Thus, `mixup` and additive noise have a complexity of $\mathcal{O}(qr)$.

**Design choices.** For certain $\lambda$ values, the new sample $Z_l(\lambda, k), Y(\lambda, k)$ may be too far from $\mathcal{P}$. With this in mind, we explored the option of scaling down the loss function $c(\cdot, \cdot)$ by a parameter $\mu(\lambda)$ in addition to modifying the singular values. However, we tested various profiles $\mu(\lambda)$ and discovered the best models are obtained when no scaling of loss occurs, see App. C. Importantly, this means that our approach adopts a different ansatz in comparison to `mixup`. While `mixup` incorporates uncertainty into the model training using "in-between" samples and labels, our method uses the new data as if it was sampled from the true distribution. An alternative option which would be conceptually closer to `mixup` is to scale down the *large* singular values as well as the loss term. We show in App. C that this choice is also inferior to `comfort-zone`.

**The effect of `comfort-zone` on data and learning.** We generated a 2D point cloud whose intrinsic dimension is one (shown in blue, Fig. 1), and we applied different DA methods on this data. The three panels in the figure show in orange the augmented data when using additive noise, `mixup`, and `comfort-zone` with $\alpha = 1.0$ over the original point cloud colored in light blue. Injecting noise alters each point in its neighborhood, whereas `mixup` draws the points towards the center of their convex hull. In contrast, `comfort-zone` aligns the new samples along the dominant component of the original data. Notably, our approach may increase the span of training data, and thus it can improve estimation in regression as was recently shown in (Wu et al., 2020).

We argue that training on samples created with our method encourages the inherent tendency of the network to model the dominant parts of the data better. To demonstrate this phenomenon, we trained an N-BEATS (Oreshkin et al., 2019) architecture with and without `comfort-zone` on the Air Passengers dataset provided in DARTS (Herzen et al., 2022). The trained models are evaluated on the dataset modified using a 100 varying $\lambda \in [0, 1]$ values, see Fig. 2 (left). Namely, we modify the singular values of every batch in the dataset using different $\lambda$, and feed the resulting data for inference. Surprisingly, the non-augmented model (blue curve) performs *better* on the unseen modified samples, yielding the minimum at $\lambda \approx 0.5$. In comparison, the regularized network attains a qualitatively similar plot, but the MSE is lower for all $\lambda$ and the minimum is obtained for a lower value at $\lambda \approx 0.3$. This behavior was found to be consistent across several architectures and datasets, see App. B.

Inspecting an example of the data and its modifications, reveals the differences between the samples when select $\lambda$ values are used. The original sample (black) shown in Fig. 2 (right) exhibits primary extrema $S_p$ at times $t = 4, 7, 15$ and secondary extrema $S_q$ at times $t = 9, 13$ marked with black dots. The blue curve ($\lambda = 0.5$) for which the non-augmented model attained the minimum loss, maintains the primary extrema while significantly "flatenning" the secondary extrema. In comparison, the orange curve ($\lambda = 0.3$) for which the augmented model achieves the minimum loss, flattens $S_q$ completely. Finally, we observe that the $S_q$ data points are qualitatively different when $\lambda = 0.0$. From the analysis above, we conclude the following. First, the network prefers data with less small scale features; this finding is consistent with similar results on e.g., autoencoder models (Jain et al., 2021). Second, our regularization encourages this tendency by providing the model with such data, leading to improved MSE profiles. To the best of our knowledge, the above analysis is novel on deep regression models.

Notably, while it may argued that the behavior in Fig. 2 (left) is natural and intuitive as the model "simply" performs better on denoised signals, we argue differently. In particular, this plot somewhat contradicts our understanding of overfitting which occurs in high probability for tiny datasets such as Air Passengers (a single time series with 144 entries) using multiple weights network such as N-BEATS. Specifically, since the data is highly likely to be overfit by the network, we expect the MSE value to be lowest for $\lambda = 1$, and MSE value equal or higher for any $\lambda < 1$. Thus, we advocate that the above analysis may reveal a characteristic feature of regression neural networks. Our analysis is reinforced further as other datasets and architectures follow a similar pattern (App. B). Importantly, we are unaware of a similar experiment in the literature of deep regression neural networks.

## 4 ANALYSIS

**Relation to additive noise.** In what follows, we would like to answer the following question: Does applying `comfort-zone` is merely a variant of injecting additive noise? To this end, we analyze `comfort-zone` from a perturbation theory viewpoint. Specifically, we would like to understand how a random data perturbation affects the singular values of the data matrix $A \in \mathbb{R}^{q \times r}$, $q \geq r$. We denote by $\sigma_1 \geq \sigma_2 \geq \cdots \geq \sigma_r$ the singular values of $A$. The perturbed matrix and its singular values set are denoted by $\tilde{A} = A + E$ and $\{\tilde{\sigma}_j\}_{j=1}^r$, respectively. We write $\inf_2(A)$ and $|A|_2$ to denote the smallest and largest singular values of any matrix $A$. The following classical result provides an estimated bound for the perturbed singular values (Stewart, 1979; 1998).

**Theorem 1.** *Let $P$ be the orthogonal projection onto the column space of $A$. Let $P_\perp = I - P$. Then*

$$\tilde{\sigma}_j^2 = (\sigma_j + \gamma_j)^2 + \eta_j^2, \quad j = 1 \ldots, r,$$

*where $|\gamma_j| \leq |P E|_2$ and $\inf_2(P_\perp E) \leq \eta_j \leq |P_\perp E|_2$.*

Following Stewart (1979), we make two observations with respect to Thm. 1. First, if $\sigma_j \gg |E|_2$ then it dominates the bound and we have $\tilde{\sigma}_j \cong \sigma_j + \gamma_j$. Second and more important to our setting, when $\sigma_j$ is of order $|E|_2$, the term $\eta_j$ will tend to dominate. Indeed, in these cases the term $\eta_j$ *increases* the singular value $\sigma_j$. We conclude that random perturbations to $A$ tend to increase its small singular values. In contrast, `comfort-zone` typically decreases the small singular values, while leaving the large $\sigma_j$ unchanged. Thus, `comfort-zone` is in effect a complementary approach to injecting additive noise, allowing a finer control over the resulting new samples. Finally, we note that for a certain choice of hyper-parameters, our approach can be viewed as injecting noise per the above analysis. For example, taking $\rho = 0.0$ and $\lambda \sim \text{Uniform}(1.0, \alpha)$ for $\alpha > 1.0$ will increase all the singular values of $A$ by a factor of $\lambda \in [1.0, \alpha]$, where Uniform is the random uniform distribution.

**`comfort-zone` as a Vicinal Risk Minimization (VRM).** Given a cost function $c : \mathcal{Y} \times \mathcal{Y} \to \mathbb{R}^+$, the learning problem aims at minimizing the expectation of the loss $c(f(x), y)$ over the distribution $\mathcal{P}(x, y), x \in \mathcal{X}, y \in \mathcal{Y}$. A fundamental challenge, shared by most real-world scenarios, is that the true distribution of the data is unfortunately *unknown*. The alternative is to minimize over the empirical distribution of a train set $\{(x_i, y_i)\}_{i=1}^n$ given by

$$\mathrm{d}\,\mathcal{P}_{\text{emp}}(x, y) = \frac{1}{n} \sum_i \delta_{x_i}(x) \delta_{y_i}(y).$$

The resulting scheme is the common training procedure of modern neural networks, formally known as the Empirical Risk Minimization (ERM) (Vapnik, 1991).

While $\mathcal{P}_{\text{emp}}$ provides a basic approximation of the true $\mathcal{P}$, it was suggested (Chapelle et al., 2001) that other density estimates $\mathrm{d}\,\mathcal{P}_{\text{est}}$ that take into account the *vicinity* of $(x_i, y_i)$ should be considered. The recent `mixup` approach (Guo et al., 2019) exploits this idea by proposing a Vicinal Risk Minimization (VRM) procedure that is based on the vicinal distribution estimate $\frac{1}{n}\sum_{i,j}\delta_{\tilde{x}_{ij}(\lambda)}(x)\delta_{\tilde{y}_{ij}(\lambda)}(y)$, defined using convex combinations $\tilde{z}_{ij}(\lambda) = \lambda z_i + (1-\lambda)z_j$ for $z \in \{x, y\}$ and $\lambda \sim \text{Beta}(\alpha, \alpha)$. In this context, the main difference between `comfort-zone` and `mixup` is in the definition of vicinity as we describe below.

We denote by $\mathcal{T}(x, y)$ the tangent plane of the data manifold $\mathcal{M}$ at the point $(x, y) \in \mathcal{M} \subset \mathcal{X} \times \mathcal{Y}$. Namely, $\mathcal{T}(x, y)$ is the linear approximation of $\mathcal{M}$ at $(x, y)$. For every pair $(x, y)$, we define a new density distribution $\mathcal{P}_{\text{tan}}$ which considers all pairs $(a, b)$ in the tangent plane of $(u, v) \in \mathcal{M}$. Formally,

$$\mathrm{d}\,\mathcal{P}_{\text{tan}}(x, y) = \int_{\mathcal{M}} \int_{\mathcal{T}(u,v)} \delta_a(x)\delta_b(y)\,\mathrm{d}\,ab\,\mathrm{d}\,uv\,.$$

Then, `comfort-zone` approximates the latter expression by generating an estimate of the tangent plane $\mathcal{T}_{\text{est}}$ via `SVD`, yielding the following vicinal estimate

$$\mathrm{d}\,\mathcal{P}_{\text{est}}(x, y) = \frac{1}{n}\sum_i \frac{1}{k_i}\sum_j \delta_{x_j}(x)\delta_{y_j}(y)\,,$$

$$(x_j, y_j) \in \mathcal{T}_{\text{est}}(x_i, y_i)\,,\ \ k_i = |\mathcal{T}_{\text{est}}(x_i, y_i)|\,.$$

## 5 EXPERIMENTS

### 5.1 REGRESSION BENCHMARK DATASETS

While there is extensive work on deep regression in the vision community for e.g., object detection (Szegedy et al., 2013) and human pose estimation (Li & Chan, 2014), we aimed for an evaluation setting where data modalities different from images and text are being considered. To this end, we evaluate `comfort-zone` on regression benchmark datasets that frequently appear in the literature, see e.g., Hernández-Lobato & Adams (2015). The datasets include Diabetes listing 442 patients with 10 feature variables (Efron et al., 2004); Concrete describes 1030 instances of the actual concrete strength using 8 features (Yeh, 1998); Energy details the energy efficiency of 768 building shapes using 8 variables (Tsanas & Xifara, 2012); and Wine which consists of 1599 red wine instances with 11 features (Cortez et al., 2009). The output of Diabetes, Concrete and Wine has one feature, whereas Energy has two features. We perform min-max normalization to all datasets, and we remove it during model testing.

The baseline architecture we consider is a residual network (`ResNet`) (He et al., 2016). `ResNet` models are typically overparameterized, and thus they serve as a good baseline to explore DA effects. We use fully connected layers in the residual block instead of convolutions, employing 18 (`ResNet18`) and 34 (`ResNet34`) residual layers followed by a linear layer. During training, data is split to $70\%, 10\%$ and $20\%$ for the train, validation and test sets, respectively. Each model is trained 20 times with random splits, and it is trained for 40 epochs using a hidden size of 100 and a batch size of 16. We employ an Adam optimizer, with $0.0001$ weight decay and an initial learning rate of $0.001$, and we reduce it by half with a patience of 3 based on the validation loss. The loss is MSE, and we infer over the models which yield the best loss on the validation set when averaged over 20 runs.

We compare the baseline (`ERM`) to `mixup` (Guo et al., 2019), additive uniform noise (`UN`), and `comfort-zone` (`CZ`). The results are detailed in Tab. 1 using the metrics: root MSE (RMSE), mean absolute percentage error (MAPE), and R2 (Makridakis & Hibon, 2000). In `mixup`, we follow the authors guidelines and use $\alpha \in [0.1, 0.4]$, and with the additive noise we use a scale of $0.1$. We apply `comfort-zone` at the input level, and we perform a grid search over $\alpha \in \{0.5, 1.0, 2.0\}$, and $\rho \in \{0.97, 0.98\}$. Our results show that more depth yields inferior results, which may be related to the network size w.r.t dataset size. Injecting uniform noise to the data generally under-performs w.r.t the baseline. In contrast, `mixup` improves `ResNet18` generalization, and its results on `ResNet34` are not as effective. Our approach always yields improved test errors, except for Concrete with

Table 1: Test errors on regression benchmarks using `ResNet18` and `ResNet34` architectures.

| | Diabetes, `ResNet18` | | | Diabetes, `ResNet34` | | |
|---|---|---|---|---|---|---|
| | RMSE↓ | MAPE↓ | R2↑ | RMSE↓ | MAPE↓ | R2↑ |
| ERM | $76.00 \pm 9.28$ | $56.00 \pm 7.64$ | $-0.11 \pm 0.27$ | $82.24 \pm 9.26$ | $61.16 \pm 10.64$ | $-0.29 \pm 0.29$ |
| mixup | $73.76 \pm 8.88$ | $53.55 \pm 8.71$ | $-0.04 \pm 0.25$ | $81.12 \pm 8.63$ | $58.53 \pm 7.64$ | $-0.26 \pm 0.27$ |
| UN | $76.88 \pm 6.53$ | $58.19 \pm 4.75$ | $-0.12 \pm 0.19$ | $86.14 \pm 18.69$ | $61.14 \pm 8.26$ | $-0.47 \pm 0.79$ |
| CZ | $\mathbf{71.79 \pm 5.26}$ | $\mathbf{51.80 \pm 7.09}$ | $\mathbf{0.02 \pm 0.14}$ | $\mathbf{77.69 \pm 5.58}$ | $\mathbf{56.63 \pm 6.23}$ | $\mathbf{-0.15 \pm 0.17}$ |
| | Concrete, `ResNet18` | | | Concrete, `ResNet34` | | |
| ERM | $10.58 \pm 1.72$ | $29.52 \pm 3.51$ | $0.55 \pm 0.17$ | $\mathbf{11.87 \pm 0.74}$ | $36.65 \pm 3.50$ | $0.45 \pm 0.07$ |
| mixup | $10.21 \pm 0.77$ | $31.14 \pm 2.68$ | $0.59 \pm 0.06$ | $12.41 \pm 1.27$ | $36.68 \pm 4.27$ | $0.40 \pm 0.12$ |
| UN | $11.19 \pm 0.85$ | $35.65 \pm 4.07$ | $0.51 \pm 0.08$ | $14.33 \pm 4.00$ | $45.25 \pm 11.87$ | $0.14 \pm 0.60$ |
| CZ | $\mathbf{9.87 \pm 0.61}$ | $\mathbf{28.35 \pm 3.47}$ | $\mathbf{0.62 \pm 0.05}$ | $11.91 \pm 0.74$ | $\mathbf{35.85 \pm 3.64}$ | $0.45 \pm 0.07$ |
| | Energy, `ResNet18` | | | Energy, `ResNet34` | | |
| ERM | $3.70 \pm 0.29$ | $12.93 \pm 1.33$ | $0.86 \pm 0.02$ | $4.35 \pm 0.30$ | $15.75 \pm 1.68$ | $0.81 \pm 0.03$ |
| mixup | $3.78 \pm 0.27$ | $12.88 \pm 0.95$ | $0.86 \pm 0.02$ | $4.72 \pm 0.39$ | $17.08 \pm 1.69$ | $0.78 \pm 0.04$ |
| UN | $3.83 \pm 0.24$ | $13.69 \pm 1.03$ | $0.85 \pm 0.02$ | $5.98 \pm 3.09$ | $22.85 \pm 12.41$ | $0.55 \pm 0.58$ |
| CZ | $\mathbf{3.61 \pm 0.16}$ | $\mathbf{12.25 \pm 0.70}$ | $\mathbf{0.87 \pm 0.01}$ | $\mathbf{4.29 \pm 0.33}$ | $\mathbf{15.23 \pm 1.50}$ | $\mathbf{0.81 \pm 0.03}$ |
| | Wine, `ResNet18` | | | Wine, `ResNet34` | | |
| ERM | $0.68 \pm 0.03$ | $9.89 \pm 0.49$ | $0.28 \pm 0.05$ | $0.75 \pm 0.09$ | $10.66 \pm 0.82$ | $0.13 \pm 0.26$ |
| mixup | $0.67 \pm 0.04$ | $9.52 \pm 0.53$ | $0.31 \pm 0.08$ | $0.74 \pm 0.05$ | $10.48 \pm 0.59$ | $0.16 \pm 0.12$ |
| UN | $0.69 \pm 0.03$ | $9.96 \pm 0.51$ | $0.28 \pm 0.05$ | $0.73 \pm 0.03$ | $10.58 \pm 0.55$ | $0.18 \pm 0.07$ |
| CZ | $\mathbf{0.65 \pm 0.03}$ | $\mathbf{9.33 \pm 0.42}$ | $\mathbf{0.34 \pm 0.06}$ | $\mathbf{0.72 \pm 0.05}$ | $\mathbf{10.27 \pm 0.71}$ | $\mathbf{0.20 \pm 0.11}$ |

34 layers by a small RMSE margin. Further, our method achieves the best results on all datasets in comparison to all other DA baselines. We also find that `comfort-zone` reduces the standard deviation for almost all datasets and metrics.

## 5.2 Time series forecasting (TSF)

**Small-scale TSF.** Forecasting time series data is one of the fundamental regression tasks in machine learning. We test `comfort-zone` using the DARTS (Herzen et al., 2022) time series forecasting framework, which supports several TSF methods and datasets. Specifically, we consider the DARTS implementations of `RNN`, `TCN` (Bai et al., 2018), `TRANSFORMER` (Vaswani et al., 2017), and `N-BEATS` (Oreshkin et al., 2019). The datasets are mostly univariate, i.e., the time series samples are one-dimensional. In our experiments, we perform a min-max normalization to the data, and we convert it to a single-precision floating point representation. Importantly, DARTS datasets are *small*, ranging from $\approx 100$ samples to $\approx 3000$ samples in total. This regime of small training sets is expected to benefit the most from DA techniques such as ours.

The data is split to approximately $80\%$ for training and $20\%$ for testing. Unless otherwise noted, covariates such as hour-of-day are not used (Salinas et al., 2020). We train for 300 epochs, using a batch size of 32 and an Adam optimizer with a learning rate 0.001 and no scheduling. In all cases, the training loss is mean squared error (MSE). The specific input and output tensor sizes depend on the dataset, and we provide this information in App. D. For reproducibility and to reduce variability, we train each model on the same hundred seeds $\{0, \dots, 99\}$. During inference, we evaluate the trained models using the root mean square error (RMSE), mean absolute percentage error (MAPE), and R2 measures, see e.g., (Makridakis & Hibon, 2000). We report the average measures and their standard deviation over the seed set. In our experiments, we compared the effect of `comfort-zone` in relation to the baseline model (`ERM`), and to the baseline augmented with the DA approaches `mixup`, additive noise (`UN`), and `comfort-zone` (`CZ`). Following the evaluation protocol proposed in the original `mixup` paper (Zhang et al., 2017), we evaluate the dependence of different DA methods on the choice of hyper-parameters. To this end, we fix the hyper-parameters for all DA baselines. We used $\alpha = 0.4$ for `mixup`, a scale of 0.1 for `UN`, and $\alpha = 0.2$ and $\rho = 0.9$ for `CZ` in all cases. For `comfort-zone`, we take the best result out of the original data and noise-injected data. The hyper-parameters were chosen using a basic grid test, taking the parameters which yield the best average error across DARTS datasets.

Table 2: Test errors of small-scale time series forecasting datasets from DARTS. Each dataset is trained on generic `N-BEATS`, and it is augmented using `comfort-zone` and other DA approaches.

| | Air Passengers | | | Australia Beer | | |
|---|---|---|---|---|---|---|
| | RMSE$\downarrow_{\times 10^{-2}}$ | MAPE$\downarrow$ | R2$\uparrow$ | RMSE$\downarrow_{\times 10^{-2}}$ | MAPE$\downarrow$ | R2$\uparrow$ |
| ERM | $5.66 \pm 1.2$ | $8.30 \pm 1.8$ | $0.49 \pm 0.2$ | $3.27 \pm 0.8$ | $4.79 \pm 1.2$ | $0.91 \pm 0.0$ |
| mixup | $5.24 \pm 1.1$ | $7.73 \pm 1.6$ | $0.57 \pm 0.2$ | $2.97 \pm 0.6$ | $4.60 \pm 1.0$ | $0.92 \pm 0.0$ |
| UN | $3.64 \pm 1.0$ | $5.54 \pm 1.6$ | $0.78 \pm 0.1$ | $3.72 \pm 0.9$ | $5.24 \pm 1.4$ | $0.88 \pm 0.1$ |
| CZ | $\mathbf{3.56 \pm 0.9}$ | $\mathbf{5.43 \pm 1.4}$ | $\mathbf{0.79 \pm 0.1}$ | $\mathbf{2.96 \pm 0.6}$ | $\mathbf{4.51 \pm 0.9}$ | $\mathbf{0.92 \pm 0.0}$ |

| | US Gasoline | | | Sunspots | | |
|---|---|---|---|---|---|---|
| | RMSE$\downarrow_{\times 10^{-2}}$ | MAPE$\downarrow$ | R2$\uparrow$ | RMSE$\downarrow_{\times 10^{-2}}$ | MAPE$\downarrow$ | R2$\uparrow$ |
| ERM | $6.33 \pm 0.5$ | $6.64 \pm 0.5$ | $-0.11 \pm 0.2$ | $6.17 \pm 0.9$ | $27.45 \pm 3.2$ | $-0.68 \pm 0.5$ |
| mixup | $6.40 \pm 0.5$ | $6.73 \pm 0.5$ | $-0.13 \pm 0.2$ | $6.05 \pm 0.8$ | $\mathbf{26.17 \pm 2.7}$ | $-0.61 \pm 0.4$ |
| UN | $6.49 \pm 0.6$ | $6.68 \pm 0.6$ | $-0.16 \pm 0.2$ | $\mathbf{6.02 \pm 0.7}$ | $27.59 \pm 3.5$ | $\mathbf{-0.59 \pm 0.4}$ |
| CZ | $\mathbf{6.25 \pm 0.6}$ | $\mathbf{6.59 \pm 0.6}$ | $\mathbf{-0.08 \pm 0.2}$ | $6.09 \pm 0.9$ | $27.20 \pm 3.4$ | $-0.64 \pm 0.5$ |

Tab. 2 shows the statistics and results for the univariate datasets Air Passengers, Australia Beer, US Gasoline and Sunspots provided in DARTS. These datasets are trained on the baseline `N-BEATS` architecture whose time series forecasting capabilities are considered state-of-the-art (Oreshkin et al., 2019), and then trained again on baseline with DA. We observe a consistent behavior where `comfort-zone` improves the generalization error compared to `ERM` and usually reduces the standard deviation for all datasets. Further, `comfort-zone` beats all other methods, except on Sunspots where the best results for MAPE are attained by `mixup`, and for RMSE and R2 by additive noise. In addition to Tab. 2, we show in Tab. 9 an extended evaluation, showing the results on DARTS datasets on the baseline architectures `RNN`, `TCN` and `TRANSFORMER`, and on their DA augmented versions. In this extended setting, we observe that `TCN` and `TRANSFORMER` benefit from our DA for all datasets, whereas `RNN` yields mixed results with `comfort-zone`. Furthermore, the standard deviation typically becomes smaller with our DA in comparison to the baseline. Additive noise and `mixup` somewhat depend on the architecture and dataset, where in some cases the generalization improves and in others, deteriorates. Notably, the best overall results (marked in blue) for each dataset were almost always obtained with `comfort-zone`. The only exception was for RMSE and R2 metrics for US Gasoline, where `comfort-zone` yields the best MAPE results, and second best RMSE and R2 (marked in red). Finally, we note that while our `CZ` on Sunspot with `N-BEATS` in Tab. 2 did not yield the best estimates in comparison to other DA techniques, the setting of `TCN` with `CZ` attains the best *overall* results for Sunspots, see Tab. 9.

**Large-scale TSF.** To further evaluate our approach in the context of forecasting, we consider larger-scale benchmark datasets. Electricity contains the hourly electricity consumption of 370 customers for a total of $\approx 9.7$M samples, and Traffic includes the hourly occupancy rate of 963 car lanes of San Francisco bay area freeways for a total of $\approx 10.1$M samples. Both datasets appear frequently in the forecasting literature, e.g., (Salinas et al., 2020; Oreshkin et al., 2019). We incorporate our `comfort-zone` into the `N-BEATS` framework which includes ensembling during inference. Following Oreshkin et al. (2019), we train the generic and interpretable models with and without DA, and we use a total of 180 models for evaluation. These models arise from using different metrics, different horizon lengths, and different initialization. Finally, different data splits are considered. Using the original code repository of the authors, we approximately recover their results. We refer to (Oreshkin et al., 2019) for the full details regarding the evaluation protocol and testing setup. We perform a grid search over $\alpha \in \{0.1, \ldots, 1.0\}$ and $\rho \in \{0.8, 0.85, 0.9\}$ yielding a total of 30 models per dataset, architecture and split. Many hyper-parameter combinations lead to an improvement in the normalized deviation (ND) test error. Tab. 3 shows the ensemble median results of the generic (`N-BEATS-G`) and interpretable (`N-BEATS-I`) baselines, as well as our results. While `comfort-zone` improves both datasets, we observe that the generic net benefits relatively more from our DA in comparison to its interpretable version.

We further extend our evaluation on large-scale TSF tasks where we consider the datasets ETTm$_2$, Exchange, weather and ILI in the challenging setting of long horizon forecasting benchmark including

Table 3: Test errors for `N-BEATS` architectures generic (`G`) and interpretable (`I`) trained with and without `comfort-zone` on Electricity and Traffic datasets for different train-test splits.

| | Electricity | | | Traffic | | |
|---|---|---|---|---|---|---|
| | 2014-09-01 | 2014-03-31 | last 7 days | 2008-06-15 | 2008-01-14 | last 7 days |
| `N-BEATS-G` | 0.065 | 0.066 | 0.179 | 0.129 | 0.231 | **0.115** |
| `N-BEATS-G + CZ` | **0.062** | **0.062** | **0.166** | **0.118** | **0.231** | 0.116 |
| `N-BEATS-I` | 0.072 | 0.071 | 0.179 | 0.122 | 0.234 | 0.118 |
| `N-BEATS-I + CZ` | **0.070** | **0.068** | **0.174** | **0.12** | **0.231** | **0.116** |

forecasts of lengths $96, 192$ and $336$. We refer the reader to Wu et al. (2021) where the datasets are described as well as the benchmark setting. For a baseline, we consider the generic version of `N-BEATS`, trained in `ERM` and augmented with `CZ`. We performed a grid search for `comfort-zone` using $\rho \in \{0.85, 0.90, 0.95\}$, and $\alpha \in \{0.1, \ldots, 1.0\}$. Many hyper-parameter combinations lead to improved results over the baseline, and we report the best results of our approach per horizon. Tab. 4 reports the MSE and mean absolute error (MAE) metrics of this experiment. In all cases except for ETTm$_2$ with horizon 96, `CZ` improves generalization and yields better error metrics.

Table 4: Long horizon time series forecasting results.

| | | ETTm$_2$ | | | Exchange | | | Weather | | | ILI | | |
|---|---|---|---|---|---|---|---|---|---|---|---|---|---|
| | | 96 | 192 | 336 | 96 | 192 | 336 | 96 | 192 | 336 | 96 | 192 | 336 |
| ERM | MSE | **0.173** | 0.250 | 0.301 | 0.101 | 0.157 | 0.320 | 0.169 | 0.220 | 0.284 | 1.830 | 2.074 | 2.329 |
| | MAE | **0.257** | 0.312 | 0.354 | 0.216 | 0.288 | 0.423 | 0.205 | 0.250 | 0.302 | 0.883 | 0.961 | 1.062 |
| CZ | MSE | 0.175 | **0.242** | **0.297** | **0.078** | **0.156** | **0.298** | **0.164** | **0.217** | **0.275** | **1.798** | **1.997** | **2.306** |
| | MAE | 0.259 | **0.310** | **0.349** | **0.195** | **0.287** | 0.401 | **0.201** | **0.247** | **0.293** | **0.854** | **0.939** | **1.039** |

## 6 DISCUSSION

We have proposed `comfort-zone`, a data-driven method for data augmentation of regression tasks. We showed that `comfort-zone` supports the network tendency of representing dominant components of its input signals by creating virtual examples sampled from the tangent planes of the original train set. Implementing `comfort-zone` is straightforward, and it admits a fully differentiable as well as a simpler non-differentiable versions. Throughout an extensive evaluation, we have shown that `comfort-zone` improves the generalization error of neural models on time series forecasting datasets and regression benchmarks. In addition, `comfort-zone` obtains better results when compared to a few data augmentation baselines, while reducing the standard deviation of the model ensemble.

When inspecting the effect of the hyper-parameters $\alpha$ and $\rho$, we observe that for small datasets the results improve as $\alpha$ increases, and for medium datasets the results are stable or deteriorate for increasing $\alpha$. Further, larger neural models (`ResNet34`) were less affected by changes in $\alpha$ in comparison to smaller models (`ResNet18`). We identify that choosing the value of $\rho$ depends on the intrinsic features of the dataset. In general, higher $\rho$ is preferable when the intrinsic dimension of the data is higher. However, our understanding of the interplay between the hyper-parameters and model behavior is still somewhat limited. The time complexity of `comfort-zone` is governed by the `SVD` calculation, which may be restrictive for large train batches.

There are several exciting avenues for future exploration. First, is there a fundamental link between the vicinal distribution employed and the learned representation? While several existing works suggest that *linearity* yields better models, the model dependency on the specific definition of vicinity is still not well understood. Second, can similar methods be useful in classification tasks? The adaptation of `comfort-zone` to classification is straightforward, however, several design choices which were tuned for regression may require change in a classification setting.

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

## A  A FULLY DIFFERENTIABLE COMFORT-ZONE

In Sec. 3 and Fig. 1 we discuss a potential implementation of our method at the input level, i.e., for $l = 0$. However, this approach is not suitable for the latent version. Indeed, identifying the indices of singular values which should be scaled by $\lambda$ as was proposed in Sec. 3 is not a differentiable action. Specifically, the use of numpy.where() does not allow for end-to-end learning, and it should be replaced. Fortunately, PyTorch allows for differentiable index selecting from a tensor, thus by using this feature we can separate the singular values we wish to scale from those we wish to keep as is. We separate the s, the singular values vector, into two vectors, scale the desired singular values and then concatenate the vectors, a differentiable operation in itself, to recreate s. For completeness, we provide the pseudocode for the fully differentiable scale_down function in Fig. 3.

```
def scale_down(Z, lam, rho):
    U, s, Vt = torch.linalg.svd(Z)
    cumperc = torch.cumsum(s) / torch.sum(s)
    Jts = torch.nonzero(cumperc > rho, True)[0]
    Jbs = torch.nonzero(cumperc <= rho, True)[0]
    ts = s.index_select(dim=0, index=Jts)
    bs = s.index_select(dim=0, index=Jbs)
    s = torch.cat((ts, bs * lam))
    Z = U @ torch.diag(s) @ Vt
```

Figure 3: A differentiable version of the scale_down function.

## B  SEQUENTIAL MODELS CAPTURE DOMINANT COMPONENTS OF DATA BETTER

Following the discussion in Sec. 3, we verify empirically that neural networks model the dominant parts of their data better. We repeat the experiment in Fig. 2 in the main text using several datasets and architectures. Every pair of dataset and architecture are evaluated on the dataset whose singular values are modified using varying values of $\lambda$. The results are presented in Fig. 4 where solid lines represent the non-regularized version, and dashed lines are associated with models trained with our DA. In *all* cases we observe a similar qualitative behavior as we reported in Sec. 3. In particular, the highest MSE values are obtained for both the baseline and regularized models for $\lambda = 1$, i.e., when the data is unchanged. Further, the model attain improved error measures as $\lambda$ decreases, where the error profile is similar for the baseline and regularized models. Based on these results, we deduce that sequential models prefer to represent and compute the dominant components of data.

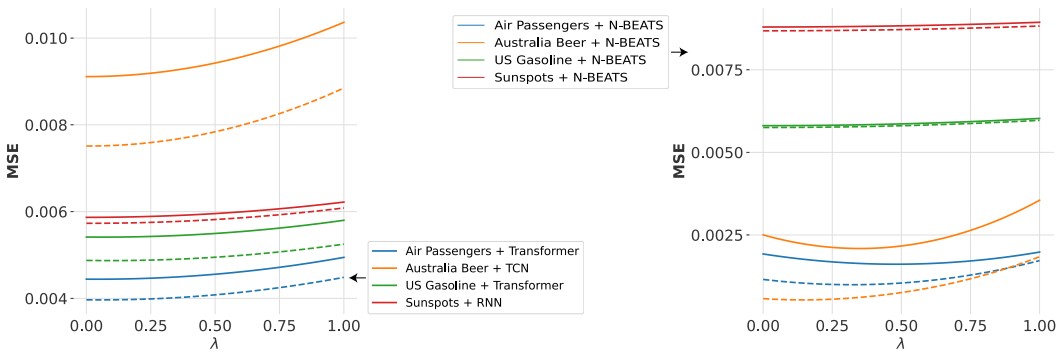

Figure 4: We reproduce Fig. 2 for several architectures and datasets. In all cases the models achieve better error measures for the modified data, whether it appeared during training or not.

## C  ABLATION STUDY

To motivate the specific design choices in `comfort-zone`, we run an ablation study over different design settings. The first hyper-parameter we consider is $\mu(\lambda)$ which is used to scale the cost function during training. Our experiments show that $\mu(\lambda) = 1$, i.e., no scaling, leads to the best results, and we report for profiles $\mu(\lambda) \in \{1, \lambda, \lambda^2\}$. The second hyper-parameter marks whether to scale down the small or large singular values. In `comfort-zone` we always scale the small singular values. The third hyper-parameter deals with modifying the samples at the input level or in the latent space. The results are given in Tab. 5. The ablation study is performed on the Concrete regression dataset using `ResNet18` architecture, and Australia Beer dataset using `N-BEATS` architecture. For both datasets, scaling down the singular values at the input level and with no scaling to the loss function leads to the best test measures. Further, the latent version of `comfort-zone` yields the second best results. Finally, scaling down the large singular values and the loss function was beneficial for Australia Beer, but resulted in poor measures on Concrete.

Table 5: Ablation study of `comfort-zone` over different loss scaling profiles $\mu(\lambda)$, scaling down the small or large singular values, and modifying data at the input or latent levels.

| Data | #samples | #feats | mode | $\mu(\lambda)$ | scale | RMSE$\downarrow$ | MAPE$\downarrow$ | R2$\uparrow$ |
|---|---|---|---|---|---|---|---|---|
| Concrete | 1030 | 8 | input | 1 | small | $\mathbf{9.87 \pm 0.61}$ | $\mathbf{28.35 \pm 3.47}$ | $\mathbf{0.62 \pm 0.05}$ |
| | | | input | $\lambda$ | small | $10.60 \pm 1.04$ | $31.35 \pm 3.20$ | $0.56 \pm 0.09$ |
| | | | input | $\lambda^2$ | small | $11.29 \pm 0.80$ | $33.17 \pm 3.05$ | $0.50 \pm 0.07$ |
| | | | input | $\lambda$ | large | $22.89 \pm 4.33$ | $61.62 \pm 13.97$ | $-1.10 \pm 0.78$ |
| | | | latent | 1 | small | $10.20 \pm 0.71$ | $30.56 \pm 3.56$ | $0.59 \pm 0.06$ |
| Australia Beer | 176 | 1 | input | 1 | small | $\mathbf{0.030 \pm 0.006}$ | $\mathbf{4.510 \pm 0.927}$ | $\mathbf{0.924 \pm 0.031}$ |
| | | | input | $\lambda$ | small | $0.034 \pm 0.009$ | $4.989 \pm 1.427$ | $0.899 \pm 0.050$ |
| | | | input | $\lambda^2$ | small | $0.035 \pm 0.008$ | $5.150 \pm 1.314$ | $0.890 \pm 0.051$ |
| | | | input | $\lambda$ | large | $0.031 \pm 0.005$ | $4.799 \pm 0.892$ | $0.918 \pm 0.028$ |
| | | | latent | 1 | small | $0.032 \pm 0.009$ | $4.732 \pm 1.324$ | $0.908 \pm 0.050$ |

## D  TIME SERIES FORECASTING EVALUATION DETAILS

Tab. 6 shows several training parameters related to the time series forecasting experiments on DARTS datasets reported in Sec. 5. The split column specifies the point in time from which we split the data to train and test sets. Then #in and #out represent the series length for the input and output, respectively. Finally, #pred is the length of series predicted during model evaluation.

Table 6: Training parameters of DARTS datasets.

| Data | split | #in | #out | #pred |
|---|---|---|---|---|
| Air Passengers | 1958-12-01 | 12 | 6 | 6 |
| Australia Beer | 2002-06-01 | 12 | 6 | 6 |
| US Gasoline | 2015-01-01 | 125 | 18 | 18 |
| Sunspots | 1940-10-01 | 125 | 18 | 18 |

## E  EFFECTS OF ADDITIVE NOISE IN COMBINATION WITH COMFORT-ZONE

In our experiments, we tested DARTS time series datasets using `comfort-zone` (CZ) and additive noise UN, together and separately. We reported the results achieving the best metrics with or without UN. For the sake of completeness, we add in Tab. 7 a comparison of the results for each of our DARTS datasets, when training with CZ with and without UN. Typically, there is some improvement when using UN alongside CZ, but this is not always the case, e.g., for Australia Beer and `N-BEATS` where CZ alone had better results. For each dataset and architecture we also add the baseline without any DA, which shows that generally CZ improves on the baseline with our without UN. We believe additive noise is helpful in this test scenario since the DARTS datasets are extremely small, and thus our data augmentation does not necessarily span a wide enough regime.

Table 7: Test errors on several sequential neural architectures on the small-scale time series forecasting datasets from DARTS. Each architecture is trained with `comfort-zone` and either with (`CZ+UN`) or without additive uniform noise (`CZ`).

| Data | #samples | Method | RMSE↓ ×10⁻² | MAPE↓ | R2↑ |
|---|---|---|---|---|---|
| Air Passengers | 144 | N-BEATS | $5.66 \pm 1.2$ | $8.30 \pm 1.8$ | $0.49 \pm 0.2$ |
| | | N-BEATS + CZ | $4.78 \pm 1.0$ | $7.18 \pm 1.7$ | $0.64 \pm 0.1$ |
| | | N-BEATS + CZ + UN | $\mathbf{3.63 \pm 0.9}$ | $\mathbf{5.57 \pm 1.5}$ | $\mathbf{0.79 \pm 0.1}$ |
| | | TRANSFORMER | $5.23 \pm 1.0$ | $7.25 \pm 1.8$ | $0.59 \pm 0.2$ |
| | | TRANSFORMER + CZ | $5.02 \pm 1.2$ | $6.96 \pm 1.8$ | $0.60 \pm 0.2$ |
| | | TRANSFORMER + CZ + UN | $\mathbf{4.35 \pm 1.0}$ | $\mathbf{6.20 \pm 1.5}$ | $\mathbf{0.70 \pm 0.2}$ |
| Australia Beer | 176 | N-BEATS | $3.27 \pm 0.8$ | $4.79 \pm 1.2$ | $0.91 \pm 0.03$ |
| | | N-BEATS + CZ | $\mathbf{2.90 \pm 0.7}$ | $\mathbf{4.46 \pm 1.0}$ | $\mathbf{0.92 \pm 0.03}$ |
| | | N-BEATS + CZ + UN | $3.21 \pm 0.7$ | $4.66 \pm 1.1$ | $0.91 \pm 0.04$ |
| | | TRANSFORMER | $4.94 \pm 0.9$ | $6.51 \pm 1.4$ | $0.80 \pm 0.08$ |
| | | TRANSFORMER + CZ | $5.27 \pm 1.2$ | $7.07 \pm 2.0$ | $0.76 \pm 0.11$ |
| | | TRANSFORMER + CZ + UN | $\mathbf{4.55 \pm 1.0}$ | $\mathbf{5.94 \pm 1.4}$ | $\mathbf{0.82 \pm 0.08}$ |
| US Gasoline | 1578 | N-BEATS | $6.33 \pm 0.5$ | $6.64 \pm 0.5$ | $-0.11 \pm 0.2$ |
| | | N-BEATS + CZ | $\mathbf{6.30 \pm 0.6}$ | $\mathbf{6.65 \pm 0.6}$ | $\mathbf{-0.10 \pm 0.2}$ |
| | | N-BEATS + CZ + UN | $6.35 \pm 0.6$ | $6.61 \pm 0.6$ | $-0.12 \pm 0.2$ |
| | | TRANSFORMER | $\mathbf{6.07 \pm 0.4}$ | $\mathbf{6.48 \pm 0.4}$ | $\mathbf{-0.01 \pm 0.1}$ |
| | | TRANSFORMER + CZ | $6.15 \pm 0.4$ | $6.52 \pm 0.4$ | $-0.04 \pm 0.2$ |
| | | TRANSFORMER + CZ + UN | $6.15 \pm 0.4$ | $6.44 \pm 0.4$ | $-0.04 \pm 0.2$ |
| Sunspots | 2820 | N-BEATS | $6.17 \pm 0.9$ | $\mathbf{27.45 \pm 3.2}$ | $-0.68 \pm 0.5$ |
| | | N-BEATS + CZ | $6.61 \pm 0.9$ | $27.51 \pm 3.5$ | $-0.68 \pm 0.5$ |
| | | N-BEATS + CZ + UN | $\mathbf{6.03 \pm 0.7}$ | $27.71 \pm 3.5$ | $\mathbf{-0.60 \pm 0.4}$ |
| | | TRANSFORMER | $6.40 \pm 0.9$ | $27.16 \pm 2.6$ | $-0.81 \pm 0.6$ |
| | | TRANSFORMER + CZ | $6.38 \pm 0.6$ | $27.07 \pm 2.6$ | $-0.79 \pm 0.5$ |
| | | TRANSFORMER + CZ + UN | $\mathbf{5.88 \pm 0.6}$ | $\mathbf{26.86 \pm 2.5}$ | $\mathbf{-0.51 \pm 0.3}$ |

## F    TIME CONSUMPTION OF APPLYING `COMFORT-ZONE`

We add in Tab. 8 the timings of applying `CZ` on a batch of the different time series datasets. We measured these timings by applying `CZ` on the batch 1000 times, measuring the entire time length then dividing the total time by 1000 to get the average time of a single application.

Table 8: Running times of a single application of our method for several datasets.

| Data | Data Size | Method timing×10⁻⁴ (ms) |
|---|---|---|
| Air Passengers | $32 \times 18$ | 5.168 |
| Australia Beer | $32 \times 18$ | 5.122 |
| US Gasoline | $32 \times 143$ | 7.210 |
| Sunspots | $32 \times 143$ | 7.426 |

## G    ADDITIONAL TIME SERIES RESULTS

In addition to the results in Tab. 2, we evaluate our method on three architectures (`RNN`, `TCN` and `TRANSFORMER`), using `ERM` and `mixup`, `UN` and `CZ` for DA approaches. We report the results in Tab. 9, where we highlight in blue the best method and in red the second-best. Overall, `CZ` achieves the best results in all cases and RMSE, MAPE and R2 metrics, except of US Gasoline where `TRANSFORMER` yields better RMSE and R2 estimates.

Table 9: Test errors of several sequential neural models on the time series forecasting datasets from DARTS. Each architecture is also trained with: `mixup`, additive noise, and `comfort-zone`.

| Data | #samples | Method | RMSE↓ ×10⁻² | MAPE↓ | R2↑ |
|---|---|---|---|---|---|
| Air Passengers | 144 | RNN | $9.117 \pm 1.64$ | $14.403 \pm 2.84$ | $-0.304 \pm 0.49$ |
| | | RNN + mixup | $9.269 \pm 1.81$ | $14.643 \pm 3.18$ | $-0.355 \pm 0.53$ |
| | | RNN + UN | $8.805 \pm 1.60$ | $14.000 \pm 2.80$ | $-0.217 \pm 0.41$ |
| | | RNN + CZ | $8.773 \pm 1.28$ | $13.918 \pm 2.27$ | $-0.194 \pm 0.34$ |
| | | TCN | $9.710 \pm 4.43$ | $15.304 \pm 7.75$ | $-0.731 \pm 2.40$ |
| | | TCN + mixup | $9.634 \pm 3.72$ | $14.901 \pm 6.28$ | $-0.620 \pm 1.62$ |
| | | TCN + UN | $8.857 \pm 2.43$ | $13.686 \pm 3.75$ | $-0.281 \pm 0.72$ |
| | | TCN + CZ | $9.141 \pm 3.63$ | $14.225 \pm 6.32$ | $-0.469 \pm 2.00$ |
| | | TRANSFORMER | $5.234 \pm 1.02$ | $7.245 \pm 1.75$ | $0.568 \pm 0.17$ |
| | | TRANSFORMER + mixup | $6.321 \pm 1.64$ | $8.876 \pm 2.70$ | $0.352 \pm 0.34$ |
| | | TRANSFORMER + UN | $4.507 \pm 1.10$ | $6.456 \pm 1.72$ | $0.673 \pm 0.17$ |
| | | TRANSFORMER + CZ | $4.352 \pm 0.99$ | $6.197 \pm 1.52$ | $0.697 \pm 0.15$ |
| Australia Beer | 176 | RNN | $4.392 \pm 1.46$ | $6.169 \pm 2.08$ | $0.822 \pm 0.16$ |
| | | RNN + mixup | $5.682 \pm 2.38$ | $8.096 \pm 3.49$ | $0.684 \pm 0.28$ |
| | | RNN + UN | $6.416 \pm 1.77$ | $9.109 \pm 2.67$ | $0.631 \pm 0.23$ |
| | | RNN + CZ | $4.522 \pm 1.71$ | $6.385 \pm 2.48$ | $0.806 \pm 0.20$ |
| | | TCN | $4.058 \pm 1.85$ | $5.939 \pm 2.51$ | $0.834 \pm 0.21$ |
| | | TCN + mixup | $4.040 \pm 2.05$ | $5.775 \pm 2.74$ | $0.829 \pm 0.25$ |
| | | TCN + UN | $4.798 \pm 2.01$ | $7.220 \pm 3.05$ | $0.775 \pm 0.22$ |
| | | TCN + CZ | $3.757 \pm 1.73$ | $5.522 \pm 2.34$ | $0.858 \pm 0.19$ |
| | | TRANSFORMER | $4.939 \pm 0.89$ | $6.507 \pm 1.44$ | $0.790 \pm 0.08$ |
| | | TRANSFORMER + mixup | $4.113 \pm 0.99$ | $5.776 \pm 1.53$ | $0.851 \pm 0.07$ |
| | | TRANSFORMER + UN | $5.914 \pm 1.47$ | $7.963 \pm 2.25$ | $0.691 \pm 0.15$ |
| | | TRANSFORMER + CZ | $4.431 \pm 0.88$ | $5.730 \pm 1.26$ | $0.830 \pm 0.07$ |
| US Gasoline | 1578 | RNN | $7.146 \pm 0.48$ | $7.996 \pm 0.67$ | $-0.406 \pm 0.19$ |
| | | RNN + mixup | $7.170 \pm 0.50$ | $8.014 \pm 0.74$ | $-0.416 \pm 0.20$ |
| | | RNN + UN | $7.322 \pm 0.47$ | $8.230 \pm 0.62$ | $-0.475 \pm 0.19$ |
| | | RNN + CZ | $7.329 \pm 0.58$ | $8.244 \pm 0.78$ | $-0.481 \pm 0.23$ |
| | | TCN | $6.943 \pm 0.40$ | $7.649 \pm 0.44$ | $-0.325 \pm 0.15$ |
| | | TCN + mixup | $6.903 \pm 0.40$ | $7.611 \pm 0.42$ | $-0.310 \pm 0.15$ |
| | | TCN + UN | $6.606 \pm 0.34$ | $7.233 \pm 0.37$ | $-0.199 \pm 0.12$ |
| | | TCN + CZ | $6.737 \pm 0.37$ | $7.399 \pm 0.43$ | $-0.248 \pm 0.14$ |
| | | TRANSFORMER | $6.066 \pm 0.38$ | $6.478 \pm 0.36$ | $-0.012 \pm 0.13$ |
| | | TRANSFORMER + mixup | $6.254 \pm 0.46$ | $6.517 \pm 0.42$ | $-0.078 \pm 0.17$ |
| | | TRANSFORMER + UN | $6.187 \pm 0.43$ | $6.461 \pm 0.40$ | $-0.054 \pm 0.15$ |
| | | TRANSFORMER + CZ | $6.153 \pm 0.44$ | $6.443 \pm 0.38$ | $-0.043 \pm 0.15$ |
| Sunspots | 2820 | RNN | $5.773 \pm 0.24$ | $28.259 \pm 1.45$ | $-0.443 \pm 0.12$ |
| | | RNN + mixup | $5.797 \pm 0.25$ | $28.842 \pm 1.44$ | $-0.455 \pm 0.13$ |
| | | RNN + UN | $5.953 \pm 0.33$ | $30.169 \pm 1.94$ | $-0.537 \pm 0.17$ |
| | | RNN + CZ | $5.759 \pm 0.26$ | $28.262 \pm 1.58$ | $-0.436 \pm 0.13$ |
| | | TCN | $5.901 \pm 0.66$ | $25.455 \pm 2.06$ | $-0.524 \pm 0.35$ |
| | | TCN + UN | $6.277 \pm 0.66$ | $28.336 \pm 2.83$ | $-0.722 \pm 0.36$ |
| | | TCN + mixup | $5.936 \pm 0.52$ | $26.062 \pm 1.88$ | $-0.535 \pm 0.28$ |
| | | TCN + CZ | $5.785 \pm 0.63$ | $25.287 \pm 2.35$ | $-0.464 \pm 0.32$ |
| | | TRANSFORMER | $6.399 \pm 0.94$ | $27.161 \pm 2.57$ | $-0.808 \pm 0.55$ |
| | | TRANSFORMER + mixup | $6.230 \pm 0.83$ | $26.653 \pm 2.54$ | $-0.707 \pm 0.48$ |
| | | TRANSFORMER + UN | $5.899 \pm 0.61$ | $26.121 \pm 2.31$ | $-0.520 \pm 0.32$ |
| | | TRANSFORMER + CZ | $5.902 \pm 0.47$ | $26.890 \pm 2.44$ | $-0.515 \pm 0.25$ |

# H    RESULTS OF CZ ON CIFAR DATASETS

In Tab. 10 we demonstrate the results of applying CZ to image datasets, as well as a comparison to application of mixup and a baseline with no DA. The results were produced using the aforementioned DA methods in their manifold setup on preact-resnet18 model and CIFAR datasets, each run with three different seeds then averaged. For the manifold-mixup and baseline code we used the repository in (Lim et al., 2021), and we added the manifold CZ version mentioned in Sec. 3 on top of it. When using CZ, we applied it on the data alone, and did not incorporate the labels into the augmentation. That is, after applying CZ to a sample, its target stayed the same. The results are unfavorable towards CZ, but it's worth pointing out that mixup was designed with classification in mind, and it augments the data using the targets as well as the input data. In contrast, CZ was designed originally for regression tasks, and even though it incorporates the targets into the DA in those setups, it is less obvious to realize how to do so with the type of targets used in classification. As mentioned, the naive way we tried did not use the targets as part of the augmentation, and we think this is the main reason for the deficit in the results. We leave further exploration of this research direction to future work.

Table 10: Accuracy results on image (CIFAR) datasets.

| Data | Method | Mean Accuracy (%) |
|---|---|---|
| CIFAR10 | Baseline | $94.673 \pm 0.071$ |
| | mixup $(\alpha = 1.0)$ | $95.507 \pm 0.076$ |
| | CZ $(\alpha = 1.0,\ \rho = 0.95)$ | $94.623 \pm 0.119$ |
| CIFAR100 | Baseline | $76.130 \pm 0.164$ |
| | mixup $(\alpha = 1.0)$ | $77.943 \pm 0.039$ |
| | CZ $(\alpha = 1.0,\ \rho = 0.99)$ | $75.747 \pm 0.135$ |

