# OpenReview forum: "Comfort Zone: A Vicinal Distribution for Regression Problems"
_ICLR.cc/2023/Conference — Submitted to ICLR 2023_

### Official Review · Reviewer_tecT · 2022-10-25

**Confidence:** 3
**Correctness:** 3
**Technical Novelty And Significance:** 2
**Empirical Novelty And Significance:** 2
**Recommendation:** 5

**Clarity, Quality, Novelty And Reproducibility:**

Clarity: Clear.
Quality: The paper lacks enough experiments, but apart from this important point, is good.
Originality: Marginal.

**Strength And Weaknesses:**

Strenghts:
- The paper is easy to follow, and explains the method well.
- Compared to the baselines, in the settings the authors evaluate on, their method performs well.

Weaknesses:
- The empirical evaluation is somewhat short. For the forecasting experiments, only N-BEATS is used. Does the method also work for other datasets (the commonly used ETT, Electricity, Weather, ...), models (N-HITS, Transformer models, TCNs, LSTMs...)? What is the impact of the method on different approaches? More experiments would be needed to make the conclusions more credible.
- While the approach is ingenious, taking the SVD of a batch and making modifications to some of the singular values is part of the set of transforms certain fields will naturally apply to their input data as part of pre-processing (e.g. signal processing). There is in this sense actual, but limited novelty.
- The method claims to be domain independent but would not actually work (without modifications) for a non-regression task, contrary to Mixup.

**Summary Of The Paper:**

The authors propose a domain-independent approach inspired by mixup, which they call comfort zone. Comfort zone works by taking the SVD of the matrix formed by the concatenation of batch inputs and outputs, and then generating augmented date by reducing the scale of some subset of the singular values (the scale and number of values are parameters).

They test this approach on regression and time series forecasting benchmarks and achieve good performance relative to the baselines they have chosen.

**Summary Of The Review:**

The paper shows that attenuating some of the singular values can result in improved performance on some specific datasets. More experiments would be needed as discussed above to make those conclusions stronger. The novelty is debatable: SVD is designed so that removing singular values is natural.

---

> ### Author Response · Authors · 2022-11-17
> **Response to Reviewer tecT**
>
> We would like to thank Reviewer tecT for acknowledging the clarity of our paper and the performance of our method. Below, we address the comments raised by Reviewer tecT. Given the opportunity, we will be happy to incorporate the modifications listed below into a final revision.
>
> 1. Empirical evaluation is short
>
> Our small-scale forecasting results include additional architectures such as the transformer, RNN, and TCN. Please see Tab. 7 and 9 in the appendix. Large-scale datasets are fundamentally important in academia and industry for the development of large-scale approaches and computational pipelines. In our work, we advocate that the setting of small-to-medium scale datasets is also fundamentally important, especially because typical deep learning methodologies struggle in this regime, and thus data augmentation approaches are of high importance. Therefore, most of our empirical setup is focused on small- and medium-scale datasets, including $8$ datasets and multiple architectures. In this regime, comfort-zone is mostly effective. In addition, we added a modest evaluation on large-scale datasets, however, our empirical results show that comfort-zone is less effective in that setting.
>
> 2. SVD is applied in pre-processing
>
> Thank you for sharing this information with us. During our writing of the related work section and afterward, we searched for works that use SVD in a similar fashion to comfort-zone. While it may be that SVD is used as a pre-processing procedure in certain communities, it does not seem to be standard practice in regression tasks, to the best of our knowledge. Would Reviewer tecT agree to share with us works that perform pre-processing with SVD? We would be happy to discuss those works in the related work section and to improve the positioning of our approach.
>
>
> 3. Method is not domain-independent
>
> There is a potential misunderstanding. Our definition of domain-independent approaches as described in the abstract and the introduction are methods that do not depend on the *data modality*. That is, one can apply mixup as well as comfort-zone to image/audio/electricity usage/etc. This is in contrast to domain-dependent approaches such as image transformation which depend on the specific data at hand. Moreover, please see also our response to Reviewer 6Zsi regarding comfort-zone as a drop-in replacement for classification tasks. In general, this is possible from a practical viewpoint, although our preliminary results show inferior performance of comfort-zone in comparison to mixup on standard image classification benchmarks.
>
> 4. Novelty is debatable
>
> Could Reviewer tecT please elaborate what they mean by that sentence?

---

### Official Review · Reviewer_6Zsi · 2022-10-25

**Confidence:** 3
**Correctness:** 3
**Technical Novelty And Significance:** 3
**Empirical Novelty And Significance:** 3
**Recommendation:** 5

**Clarity, Quality, Novelty And Reproducibility:**

This paper is well-written, with good motivation and comprehensive analysis, experimental results on small/medium regression tasks are strong, but not clear how well it can be for classification tasks as well as domain adaptation applications.

**Strength And Weaknesses:**

[Strength]

1. A nice idea of applying dominant components to mixup so that the new data points can mostly preserve the statistics of true samples.
2. Author[s] provide a comprehensive analysis of its relation to additive noise and VRMs.
3. Results on small/medium regression datasets are strong.

[Weakness]

1. Could author[s] provide any insights on why it cannot be used as a drop-in replacement to standard mixup (for classification)?
2. Isn't comfort-zone encouraging stronger local manifolds around data points? Fig. 1 sounds like an overfit to the training data. How about the results used in domain adaptation, e.g. works like https://arxiv.org/abs/2001.00677?



**Summary Of The Paper:**

This paper proposes a data-driven method called comfort-zone for data augmentation of regression tasks. By producing new samples from the given ones by scaling their small singular values by random values, it incorporates the assumption that dominant components of the train set can also be viewed as true samples. Author[s] provide both non-differentiable input-level and differentiable pipeline which is applicable to any layer. Results on small and medium regression tasks and time-series forecasting show its effectiveness.

**Summary Of The Review:**

A simple drop-in approach for many data augmentation applications with comprehensive analysis. Results on regression tasks look good but need to see its performance on classification tasks and domain adaptation applications to verify its potential.

---

> ### Author Response · Authors · 2022-11-17
> **Response to Reviewer 6Zsi**
>
> We would like to thank Reviewer 6Zsi for finding our idea to be nice, and for positively commenting on our comprehensive analysis and the strength of our results. Below, we address the comments raised by Reviewer 6Zsi. Given the opportunity, we will be happy to incorporate the modifications listed below into a final revision.
>
> 1. Drop-in replacement
>
> Classification and regression differ in the domain of the output. Typically, regression problems have real-valued inputs and outputs, and thus concatenating the input and output (i.e., considering the product manifold) as we propose in comfort-zone is a reasonable approach for statistically similar (i.e., continuous) values. In comparison, outputs (labels) in classification are discrete, and then the notion of a product manifold may be less clear. In practice, one could also apply comfort-zone to classification problems. However, for the important family of classification problems arising in computer vision, where the input consists of thousands or more pixels, one may need to adapt comfort-zone as its complexity becomes limiting in comparison to alternative approaches (such as mixup). In our experiments, we also tried a basic adaptation of comfort-zone to image classification tasks. Unfortunately, our preliminary results show that comfort-zone performance is inferior to mixup. We will add a table with these results to the appendix of our revision. We leave further consideration and further investigation of this direction to future work.
>
> 2. Stronger local manifolds
>
> There seems to be a potential misunderstanding. The original point cloud (blue) in Fig. 1 represents a *noisy* sampling of the data manifold, which for that example is one-dimensional, and thus better captured with the orange point cloud obtained with comfort-zone. From this point of view, Fig. 1 does not represent an overfit to the training data, but rather, comfort-zone is able to span a distribution closer to the "true" distribution. Further, we advocate in our work that the blue point cloud is the typical case, and thus, sampling approaches such as comfort-zone are required. Please also see the response to Reviewer 5t1j regarding the relation to noise injection. Thank you for suggesting the work on domain adaptation (<https://arxiv.org/abs/2001.00677>). Including consistency constraints sounds like a great idea which we are happy to discuss in the previous work section and consider in future work.

---

> > ### Comment · Reviewer_6Zsi · 2022-12-10
> > **Post-discussion with AC and other reviewers**
> >
> > After discussing with the other reviewers, I am convinced that the empirical evaluations are not sufficient. This is a shared opinion among all reviewers. As the authors have replied, the "true" distribution sounds more like a denoised distribution from the training data, but the question here is whether noise is a regularization or useful for classification. We do not know this because it also depends on the choices of the optimizer. This could explain why mixup slightly outperforms comfort-zone, as the authors have noted. In Algorithm 1, the scale_down(.) function applies SVD to the training data corpus, which leads to non-trivial selection of the top-K eigenvectors and makes it harder to apply to larger datasets. I agree that this is difficult to fix as it is central to the paper's idea.
> >
> > I personally like the regression tasks that the authors evaluated in the paper, but I also agree with the other reviewers' thoughts. I am in a borderline position and have finalized my score as a weak-reject for now. I encourage the authors to improve their results and provide a more comprehensive evaluation in the next iteration.

---

### Official Review · Reviewer_CWda · 2022-11-03

**Confidence:** 3
**Correctness:** 3
**Technical Novelty And Significance:** 2
**Empirical Novelty And Significance:** 3
**Recommendation:** 6

**Clarity, Quality, Novelty And Reproducibility:**

This manuscript is clearly written and easy to follow. The proposed method is novel for data augmentation in regression problems. The pseudocode provided is easy to reproduce the experiments

**Strength And Weaknesses:**

Strength
+ the paper is clearly written and easy to follow. The proposed method is technically sound.
+ Extensive experiments are conducted to verify the effectiveness of proposed method.

Weakness
- Lack of comparison with SOTA data augmentation approaches
- Lack of comparison with other augmentation approaches in terms of complexity

**Summary Of The Paper:**

In this manuscript, the authors proposed a method named comfort zone which utilize the SVD to augment data for regression problems. To demonstrate the effectiveness of proposed approach, experiments across different datasets and network architecture are conducted.
~~~~~~~~~~~~~~~~~~~~~~~~~~~~~~~~~~~~~~~~~~~~~~~~~~~~~~~~~~~~~~~~~~~~~~~~~~~~~~~~~~~~~~~~~~~~~~~~~~~~
[Updates after rebuttals]

Really appreciate the authors’ efforts for addressing my concerns. After discussion with other reviewers, I finalized my score as a marginal acceptance as the proposed comfort zone is flexible and very interesting for independent domain augmentation. I also have some advises for the authors as follows.
Firstly, as replied by the authors, the proposed comfort zone has higher complexity than mixup and additive noise. Applying the SVD on large scale datasets would lead to long training time and the experimental results also show that the proposed approach can only achieve marginal improvement as pointed out by other reviewers. Secondly, I slightly agree with authors’ claim that other data augmentation methods on mentioned time series references are not applicable to the regression settings they consider. However, I would still suggest applying the proposed approach to time series regression tasks as they are very important in many real-world applications. Lastly, there is still room for improvement in terms of novelty and evaluation as mentioned by other reviewers.




**Summary Of The Review:**

Some issues need to be further clarified by the authors:

1. It is better provide the complexity comparison between the proposed one and other approaches.
2.  Besides the mix-up and UN, other advanced data augmentation approaches in regression problems should be compared. For instance, the permutation-and-jitter strategy in reference [1], decomposition-based methods [2,3] or generative models [4].

[1] Time-Series Representation Learning via Temporal and Contextual Contrasting
[2] Robusttad: Robust time series anomaly detection via decomposition and convolutional neural networks.
[3] Fast RobustSTL: Efficient and robust seasonal-trend decomposition for time series with complex patterns
[4] Time-series generative adversarial networks

3. The experiments of using proposed CZ in latent space is insufficient if the authors want to claim their approach suitable for latent space.

---

> ### Author Response · Authors · 2022-11-17
> **Response to Reviewer CWda**
>
> We would like to thank Reviewer CWda for positively commenting on the clarity of our paper and our extensive evaluation. Below, we address the comments raised by Reviewer CWda. Given the opportunity, we will be happy to incorporate the modifications listed below into a final revision.
>
> 1. Complexity comparison.
>
> Mixup samples a scalar from a random distribution, and it linearly blends two samples, whereas additive noise samples the new samples from a random distribution. In the context of the complexity analysis we provided in Sec. 3, mixup and additive noise have a complexity of $O(qr)$.
>
> 2. Comparison to SOTA DA methods
>
> There seems to be a potential misunderstanding in the focus of our paper and the problems we consider. Specifically, general regression tasks need not involve time series data. For instance, the datasets Diabetes, Concrete, Energy, and Wine we consider in Sec. 5.1 and showing their results in Tab. 1 are *not* time series datasets. Thus, the papers [1-4] mentioned by Reviewer CWda are not applicable to the general regression setting we consider. Moreover, comparing our approach to [1-4] on time series forecasting tasks (Sec. 5.2) would be an unfair comparison, as these approaches are specifically designed to handle time series data, and our method is not. The baseline methods we selected for comparison (mixup, additive noise) are the only widely used *domain-independent* DA approaches, to the best of our knowledge. We are happy to discuss papers [1-4] and include them in our related work section.
>
> 3. Insufficient experiments in latent space
>
> During our evaluation of comfort-zone, we also experimented with its latent version on all datasets and tasks. We did not include these results as they were not as good as the results we obtained by applying comfort-zone on the input. We are happy to include these results in an extended ablation, extending Tab. 5.

---

### Official Review · Reviewer_5t1j · 2022-11-04

**Confidence:** 5
**Correctness:** 4
**Technical Novelty And Significance:** 4
**Empirical Novelty And Significance:** 4
**Recommendation:** 6

**Clarity, Quality, Novelty And Reproducibility:**

The article is innovative enough, clearly presented, and of high quality. I believe there are enough details to reimplement the proposed operator in the paper.

**Strength And Weaknesses:**

**[Strength]**

- At a high level, this work has far-reaching implications for our community because: &nbsp; **(i)** Designing domain-independent data augmentation can facilitate the development of deep learning in different areas, considering that many data modalities **do not** have well-defined data augmentation operations (e.g., rotation, panning) as images do. &nbsp; **(ii)** We don't have many options for regularizing deep models on regression tasks (e.g., label smoothing can only be used for classification).
- The proposed operator ($\texttt{comfort-zone}$) is flexible because it can be applied in both input space and intermediate hidden space. Also, using SVD and scaling the "noise" or "trivial" components to generate new data is intuitive.
- $\texttt{Comfort-zone}$ yields better regularization effect than its counterpart (noise injection and mixup) on some benchmarks.

**[Weaknesses]**

- $\texttt{Comfort-zone}$ seems to be the opposite of noise injection, as it is trying to reduce the unimportant components in the data ($\lambda \in [0, 1]$). This seems to be contrary to the original purpose of data augmentation (to increase the noise in the data). What do the authors think about this?
- Performance improvement from $\texttt{comfort-zone}$ on large-scale datasets (in Sec. 5.2) seems marginal.



**Summary Of The Paper:**

This work proposes a domain-independent data augmentation operator for regression tasks: $\texttt{comfort-zone}$. It extracts the "noise" components in the data and scales them (by scaling small singular values of inputs/features). Apparently this is a new augmentation operation, and its effectiveness has been experimentally verified on some datasets.

**Summary Of The Review:**

It is necessary to explore domain-independent data augmentation as a general-purpose regularization tool. Its value is further highlighted by the applicability to regression tasks.
The proposed augmentation ($\texttt{comfort-zone}$) is well-motivated and innovative, with the only drawback that it brings very marginal improvements on large datasets.

My preliminary recommendation is to accept the article.

---

> ### Author Response · Authors · 2022-11-17
> **Response to Reviewer 5t1j**
>
> We would like to thank Reviewer 5t1j for identifying the implications of our work and its significance to the community. The points raised by Reviewer 5t1j including the extension of deep learning to different areas as well as the shortage of regularization techniques for regression tasks are some of the main motivations for us in conducting the current study. Below, we address the comments raised by Reviewer 5t1j. Given the opportunity, we will be happy to incorporate the modifications listed below into a final revision.
>
> 1. CZ is opposite to noise injection
>
> Indeed, to a large extent, comfort-zone is the opposite operator to noise injection as discussed in Sec. 4. Our view on data augmentation (DA) is that it is not necessarily meant to increase the noise in the data, but rather, DA facilitates the generation of new samples by sampling from a *vicinal distribution*, related to the true distribution of the data. If the data represents a full-rank Euclidean manifold, i.e., there are no redundant coordinates and distances can be measured in straight lines, then, noise injection is a reasonable operation, since one can sample from nearby points. However, this is rarely the case for high-dimensional data arising in machine learning tasks. In these scenarios, the underlying data manifold is low-dimensional (following the well-known manifold assumption in machine learning, see e.g., The Deep Learning Book by Goodfellow et al. 2018), and thus, noise injection provides a poor vicinal distribution estimator as it also samples from the redundant coordinates, potentially yielding out-of-distribution (OOD) samples. In comparison, comfort-zone provides a systematic approach to sample from the underlying manifold approximated linearly by SVD/PCA, thus potentially avoiding OOD samples.
>
> 2. Marginal improvement on large-scale datasets
>
> Large-scale datasets are fundamentally important in academia and industry for the development of large-scale approaches and computational pipelines. In our work, we advocate that the setting of small-to-medium scale datasets is also fundamentally important, especially because typical deep learning methodologies struggle in this regime, and thus data augmentation approaches are of high importance. Therefore, most of our empirical setup is focused on small- and medium-scale datasets, including $8$ datasets and multiple architectures. In this regime, comfort-zone is mostly effective. In addition, we added a modest evaluation on large-scale datasets, however, our empirical results show that comfort-zone is less effective in that setting.

---

### Author Response · Authors · 2022-11-27
**Addressing all the reviewers**

We want to thank the reviewers again for raising important and interesting points for us to address in your reviews. We see this process as a way to improve, and would highly appreciate your thoughts on our answers to your questions. In particular, we would be happy to clarify any remaining issues, and to resolve questions the reviewers may still have.

---

### Decision · Program_Chairs · 2023-01-20

**Decision:**

Reject

**Justification For Why Not Higher Score:**

The reviewers have common concerns about the novelty and evaluation of the paper.

**Justification For Why Not Lower Score:**

N/A

**Metareview: Summary, Strengths And Weaknesses:**

This paper proposed a data-driven method called comfort-zone for data augmentation in regression tasks. In particular, the authors utilized SVD as pre-processing, and then generated new samples by scaling the small singular values derived by SVD with random values. Experiments on regression tasks and time series forecasting tasks were conducted to show the effectiveness of the proposed method.

Strengths:
1. The paper is clearly written. The proposed method is very easy to follow.
2. The proposed method is flexible as it can work on both input space and latent feature space.
3. The model performance on small and medium regression tasks is very promising.

Weaknesses:
1. The key idea is to do data augmentation based on the SVD outputs. The novelty is thus not sufficient.
2. In addition to transformer/TCN, some strong baselines like Informer, ESTformer and Autoformer should be included.
3. The performance improvement on large-scale datasets is marginal.
4. The complexity of SVD is a concern for large-scale datasets.


**Summary Of Ac-Reviewer Meeting:**

We conducted online meetings with reviewers tecT and CWda on 9 Dec, and with 6Zsi and 5t1j on 10 Dec.

Reviewer tecT is good with the updated comparison results with Transformer and TCN. However, he still thinks stronger baselines should be included and the novelty is not sufficient.

Reviewer CWda shared that his concern on evaluating latent features has been addressed. Meanwhile, he still has the concerns about the novelty and complexity.

Reviewer 6Zsi agrees with others that the evaluation part is not sufficient although he likes the regression results. In addition, he also commented that applying SVD is a denoising process, while the noise in the data may be useful, depending on other factors like the choices of optimizer. He decided to reduce his rating from 6 to 5.